# Factors that influence older patients' participation in clinical communication within developed country hospitals and GP clinics: A systematic review of current literature

**Harry James Gaffney** [1]*, **Mohammad Hamiduzzaman**[2]

**1** College of Medicine and Public Health, Flinders University, Adelaide, Australia, **2** Faculty of Health, Southern Cross University, Gold Coast, Australia

* harry.gaffney@flinders.edu.au

## Abstract

### Background

Engaging older adults in clinical communication is an essential aspect of high-quality elder care, patient safety and satisfaction in hospitals and GP clinics. However, the factors that influence older adults' participation during their appointments with health professionals from the older patient's perspective remain under-investigated.

### Objectives

We aimed to fill this knowledge gap by reviewing research articles that have examined older patients' involvement in clinical communication. In doing so, we hope to assist healthcare professionals and institutions in developing new strategies to improve older patients' participation and engagement in clinical communication.

### Methods

A systematic review of nine databases was conducted for studies reporting identified influences on older patients' participation in clinical communication published from 2010. These studies were then subjected to thematic analysis for stratification.

### Results

Twenty-one articles with a total of 36,797 participants were included and highlighted three major themes that influenced older patients' participation in the clinical communication. The first theme identified includes *accessibility to appointments*, *support*, *health information and person-centred care*, highlighting that access to appointments, person centred care and health information significantly influences clinical communication participation. *Relevant and understandable healthcare information* identified that communication factors [i.e. tailored health information, health literacy and patient language barriers, and communication impairments] influences older patients' participation. *Older Patient perceptions of HCP*

**Funding:** The authors received no specific funding for this work.

**Competing interests:** The authors have declared that no competing interests exist.

*credibility and trustworthiness* highlighted how patient's perceptions of health professionals influence their willingness to participate in clinical communications.

## Conclusions and implications

This review demonstrates that there are several factors that contribute to insufficient or no participation of older patients in clinical communication in hospitals and GP clinics. These include accessibility to relevant and understandable health information, and the perceived health professional credibility and trustworthiness. Identifying ways to address these factors may improve patient participation, doctor-patient collaboration and overall health outcomes for older patients.

## Introduction

In recent years, health services have emphasised person-centred care. The literature points to the importance of incorporating patients' needs and preferences into communication in the clinical settings [1,2]. Effective clinical communication is paramount to person-centred care and involves a two-way interaction that generates a shared understanding of illness and treatment [3]. This effective clinical communication between healthcare professionals and their patients promotes beneficial and essential interactive outcomes, i.e. shared decision-making, bidirectional communication, empowerment and person-centred care [4–7]. These positive outcomes are associated with improved patient satisfaction and treatment outcomes [5]. Omitted care [an aspect of patient care that is omitted or delayed] is a key aspect of clinical communication that influences patient satisfaction and treatment outcomes [8,9]. Significant influencers to HCP-patient communication, engagement and subsequently missed care are patient-generated factors, e.g. patient beliefs, needs, perspectives, attitudes, health literacy and participation [10–13]. However, despite their influential role, little is known about what factors influence clinical communication with HCPs from the patient's perspective—particularly for older patients.

It is imperative to fill the research gap regarding older patients, given the ageing population worldwide. Between 2015 and 2050, the proportion of older people in the global population is expected to almost double from 12% to 22% [14]. The burden of disease associated with old age is high worldwide. Nearly 23% of the global burden of disease is attributable to the health status of older adults, with cardiovascular disease (30.3%), malignant neoplasms (15.1%), respiratory disease (9.5%), musculoskeletal problems (7.5%) and mental disorders (6.6%) accounting for the largest proportion [15]. The prevalence of chronic diseases and comorbidities in older adults is also high in developed countries such as Australia, eventually contributing 63% of the total Australian disease burden in 2011 [16].

As increasing age is one of the strongest correlates of low health literacy, it is important to address this knowledge gap [17,18]. In addition, the physical and mental health conditions of older adults can hinder effective preparation and communication in the clinical setting—increasing the likelihood of treatment failure. These include factors such as: (i) impairments in hearing, visual, cognitive function and general health [19,20]; (ii) tolerating symptoms and avoiding seeking assistance [21]; or (iii) not wanting to interrupt HCPs, feigning understanding, poor memory and poor articulation of health determinants such as pain [22–24].

At the macro level, there are several communication models designed to facilitate effective HCP-patient clinical communication [25]. The 'Activity-Passivity' and 'Guidance-

cooperation' models have historically focused on practitioner control of relationships—with patients being passive and contributing little [25,26]. These models were often described as paternalistic and focused on control of the interaction or relationship [27,28]. To counter this, Feldman-Stewart and Brundage developed a conceptual framework for patient-HCP communication that focuses on the relationships and content that occur in complex clinical environments [29]. This framework includes four components: (1) Communication goals and preparation, (2) participant attributes, (3) communication process, and (4) the environment. However, this model does not expressly consider the potential clinical participation-influencing behavioural and psychosocial differences in the older adult population compared to their younger counterparts. As health service utilisation tends to be higher in older age groups [30,31], a systematic review of the literature is important to summarise the influential factors that affect older adult patients' (OAPs) participation in clinical communication.

This systematic review aims to understand the perspectives and experiences of older patients [classified as 50 years and older in this review] in relation to the behavioural and psychosocial aspects that may affect their participation in clinical communication and influence missed care in hospitals and GP Clinics. To achieve this goal, the existing literature on this topic was summarised through a thematic analysis underpinned by models of communication in medical practices.

## Materials and methods

### Protocol and registration

This systematic review commenced on 04/01/2020 and all included article decisions were finalised on 16/06/2020. The review was conducted using a formulated protocol (S1 Protocol) (CRD42020164716) that takes into account the Population, Intervention, Comparison and Outcomes [PICO] framework [30,32] and the Preferred Reporting Items for Systematic Reviews and Meta-Analyses [PRISMA] guidelines [33]. Systematic searches were conducted in CINAHL, Cochrane, EmCare, MEDLINE, PsychINFO, Scopus, Web of Science, ProQuest and Google Scholar on 04/01/2020. The search included keywords in combination with Boolean operators, MESH terms and truncations to increase search encapsulation [See Appendix]. Grey literature was not included.

### Search strategy

An example of a search string applied to the CINAHL database to elucidate relevant articles can be found in the supporting information within the appendix (S1 Text).

**Inclusion, exclusion, and eligibility criteria.** The inclusion criteria for this review were: (i) full text, (ii) peer-reviewed, and (iii) qualitative or quantitative study articles that: (i) were written in English, (ii) included older adults as participants (however, due to the low prevalence of studies with participants exclusively over 65 years of age, articles that included all or part of patients over 50 years of age were also accepted, and data relevant to the objective were extracted), (iii) examined experiences of patient interaction and challenges in clinical practise from the perspective of OAPs, (iv) were conducted in countries with 'developed economies' according to the World Economic Situation and Prospects country classification [34], and (v) were published from 2010. The decision to include only 'developed economy' countries was due to the lack of literature on this topic coming from differently classified countries. The time limit (i.e. from 2010) was included because the term 'person-centred care' was introduced by the Institute of Medicine in 2001 [35]. This ultimately began the movement away from the biomedical model and towards the biopsychosocial model of patient-centred care and communication techniques. Despite this, articles published before 2010 often still focused

predominantly on HCP knowledge when assessing clinical communication. More recent research often reflects biopsychosocial models of interaction by considering both HCP and the patient's perspective on clinical communication. Therefore, it was important to include articles from 2010 onwards in order to (i) accurately capture contemporary communication approaches and interactions between HCPs and patients, and (ii) ensure that articles also included the patient perspective—the focus of this systematic review. Articles that did not meet any of these criteria (i.e. did not focus in whole or in part on (i) the older population, (ii) clinical communication barriers, (iii) the patient perspective, or (iv) the context in countries with 'developed economy') were excluded. In addition, health professionals were considered to be all multidisciplinary professional groups involved in the healthcare of the patient/participant from a clinical communication standpoint.

## Identification of studies

The first author (HG) independently reviewed each identified title and abstract during the literature search to determine whether the study was rejected (i.e., the full text was not reviewed because it did not meet the inclusion criteria) or classified as "maybe" (i.e., further reviewed to determine whether the study met the inclusion criteria). In these cases, the full text was obtained and read, and a decision was made by HG to exclude or potentially include the article. Full text articles were then obtained for the potentially included articles. An independent review was then conducted by both HG and a research assistant. Each reviewer decided separately whether the articles met the criteria for inclusion in the final systematic review. A meeting was then held to compare, discuss and justify the independent assessments and finally make the final selection of articles to be included in the systematic literature review.

## Critical appraisal

All included Qualitative articles were critically appraised using the Critical Appraisal Skills Programme (CASP) Qualitative Tool [36] (Table 1). The included quantitative articles were critically evaluated following the JBI Critical Appraisal Checklist for Analytical Cross-Sectional Studies (Table 2). The CASP Qualitative Tool uses ten questions that address the methodological aspect of qualitative studies [37]. These questions ask the researcher to consider whether appropriate research methods were used and whether the results are meaningful and well presented [37]. The CASP Qualitative Tool was selected due to its (i) user-friendly nature, (ii) endorsement by Cochrane and the World Health Organisation [WHO] for use with qualitative evidence, and (iii) has relatively high transparency in reporting and standards of research practise [37]. The JBI Critical Appraisal Checklist for Analytical Cross-Sectional Studies uses eight questions that address the methodological aspects of quantitative cross-sectional studies [38,39]. It was selected for this review because of its ease of use and design for use in systematic reviews [39].

## Data extraction, synthesis, and analysis

The extraction and tabulation of both the qualitative and quantitative data was carried out by the first author (HG). Once completed, the data were summarised and synthesised to identify the main findings. Only the findings that related to the objectives of the study were considered for synthesis using thematic analysis. Thematic analysis was chosen for this mixed-methods analysis because of its ability to overlap narrative summary and content analysis while clearly identifying and organising salient themes within different studies and methodologies [40]. This was done following Braun and Clarke's six stages of thematic analysis [41]. This involved the use of a reflective journal in conducting: (1) initial familiarisation with reoccurring

**Table 1. Assessment of the qualitative studies: Critical Appraisal Skills Programme (CASP).**

| Appraisal Criteria | (Aasen et al., 2011) [58] | (Black et al., 2018) [42] | (Brooks et al., 2016) [62] | (Butterworth & Campbell, 2014) [63] | (Choi et al., 2016) [43] | (Clarke et al., 2014) [44] | (Clarke et al., 2013) [45] | (Costello et al., 2012) [46] | (Dilworth et al., 2012) [47] | (Ellins & Glasby, 2014) [48] | (Evans et al., 2012) [49] | (Gerlich et al., 2012) [60] | (Gordon et al., 2018) [50] | (Pennbrant et al., 2012) [51] | (Schröder et al., 2018) [59] | (Tobiano et al., 2015) [52] | (van Ee et al., 2018) [53] | (Waterworth et al., 2017) [54] |
|---|---|---|---|---|---|---|---|---|---|---|---|---|---|---|---|---|---|---|
| CASP1 Clear statement of aims? | Yes | Yes | Yes | Yes | Yes | Yes | Yes | Yes | Yes | Yes | Yes | Yes | Yes | Yes | Yes | Yes | Yes | Yes |
| CASP2 Is a qualitative/mixed methodology appropriate? | Yes | Yes | Yes | Yes | Yes | Yes | Yes | Yes | Yes | Yes | Yes | Yes | Yes | Yes | Yes | Yes | Yes | Yes |
| Continue? Y/N | Yes | Yes | Yes | Yes | Yes | Yes | Yes | Yes | Yes | Yes | Yes | Yes | Yes | Yes | Yes | Yes | Yes | Yes |
| CASP3 Was the research design appropriate to address the aims? | Yes | Yes | Yes | Yes | Yes | Yes | Yes | Yes | Yes | Yes | Yes | Yes | Yes | Yes | Yes | Yes | Yes | Yes |
| CASP4 Was the recruitment strategy appropriate to the aims? | Yes | Yes | Yes | Yes | Yes | Yes | Yes | Yes | Unclear | Unclear | Yes | Yes | Yes | Yes | Yes | Yes | Yes | Yes |
| CASP5 Was the data collected in a way that addressed the research issue? | Yes | Yes | Yes | Yes | Yes | Yes | Yes | Yes | Yes | Yes | Yes | Yes | Yes | Yes | Yes | Yes | Yes | Yes |
| CASP6 Has the relationship between the researcher and the participants been adequately considered? | Yes | Yes | Yes | Yes | Yes | Unclear | Yes | Yes | Unclear | Yes | Yes | Yes | Unclear | Unclear | Yes | Yes | Yes | Yes |
| CASP7 Have ethical issues been taken into consideration? | Yes | Yes | Yes | Yes | Yes | Yes | Yes | Yes | Yes | Yes | Yes | Yes | Yes | Yes | Yes | Yes | Yes | Yes |
| CASP8 Was the data analysis sufficiently rigorous? | Yes | Yes | Yes | Yes | Yes | Yes | Yes | Yes | Unclear | Yes | Yes | Yes | Yes | Yes | Yes | Yes | Yes | Yes |
| CASP9 Is there a clear statement of findings? | Yes | Yes | Yes | Yes | Yes | Yes | Yes | Yes | Yes | Yes | Yes | Yes | Yes | Yes | Yes | Yes | Yes | Yes |
| CASP10 How valuable is the research? | Very | Very | Very | Very | Very | Very | Very | Very | Very | Very | Very | Very | Very | Very | Very | Very | Very | Very |
| Overall Score #/10 | 10/10 | 10/10 | 10/10 | 10/10 | 10/10 | 9/10 | 10/10 | 10/10 | 7/10 | 9/10 | 10/10 | 10/10 | 9/10 | 9/10 | 10/10 | 10/10 | 10/10 | 10/10 |

**Table 2. Assessment of the quantitative studies: JBI critical appraisal checklist for analytical cross-sectional studies.**

| Appraisal Criteria | Articles | | |
|---|---|---|---|
| | (Fortuna et al., 2017) [55] | (Foss & Hofoss, 2011) [61] | (Gibney & Moore, 2018) [56] |
| 1. Were the criteria for inclusion in the sample clearly defined? | Yes | Yes | Yes |
| 2. Were the study subjects and the setting described in detail? | Yes | Yes | Yes |
| 3. Was the exposure measured in a valid and reliable way? | Yes | Unclear | Unclear |
| 4. Were objective, standard criteria used for measurement of the condition? | Yes | Yes | Yes |
| 5. Were confounding factors identified? | Yes | Yes | No |
| 6. Were strategies to deal with confounding factors stated? | Yes | Yes | No |
| 7. Were the outcomes measured in a valid and reliable way? | Yes | Yes | Yes |
| 8. Was appropriate statistical analysis used? | Yes | Yes | Yes |
| Overall appraisal | Include | Include | Include |
| Overall Score / 8 | 8/8 | 7/8 | 5/8 |

patterns; (2) generation of initial codes based on the reoccurring patterns; (3) Code combination to develop overarching themes; (4) evaluation of overarching themes and how they support the data; (5) further analysis of how the themes contribute to an understanding of clinical communications; (6) a final dense description and selection of themes that contribute meaningfully to this understanding of the data revealed. Ultimately, the thematic analysis was supported by the aforementioned components of effective HCP-patient communication models.

# Results

## Characteristics of included studies

A total of 9356 articles were identified through a database literature search (n = 9351) and other sources, namely google scholar (n = 5) (Fig 1). After deduplication, the remaining articles (n = 3918) were screened for inclusion based on the title and abstract. Of these studies, 3126 did not meet the inclusion criteria and were excluded after title and abstract screening. The remaining studies (n = 72) were read in full and 50 were excluded because they did not meet the inclusion criteria. A total of 21 articles were included—18 qualitative (Table 3) and 3 quantitative (Table 4).

A total of 36,797 participants were included in the review, 36,420 from the quantitative studies and 377 from the qualitative studies. Articles were conducted in a variety of locations, including the UK (n = 5), US (n = 3), Australia (n = 2), Norway (n = 2), Germany (n = 2), Scotland (n = 1), Canada (n = 1), Ireland (n = 1), Sweden (n = 1), Netherlands (n = 1) and New Zealand (n = 1). One article included multiple locations in its study, including the UK, Germany and Belgium (n = 1). The age range of participants was 18–95 years, but only data from patients aged 50 years and older were used. In the 18 qualitative articles, data were collected via focus groups and retrospective semi-structured interviews either prospectively or via analysis of existing interviews (Table 3). In the three quantitative articles, cross-sectional studies were conducted (Table 2).

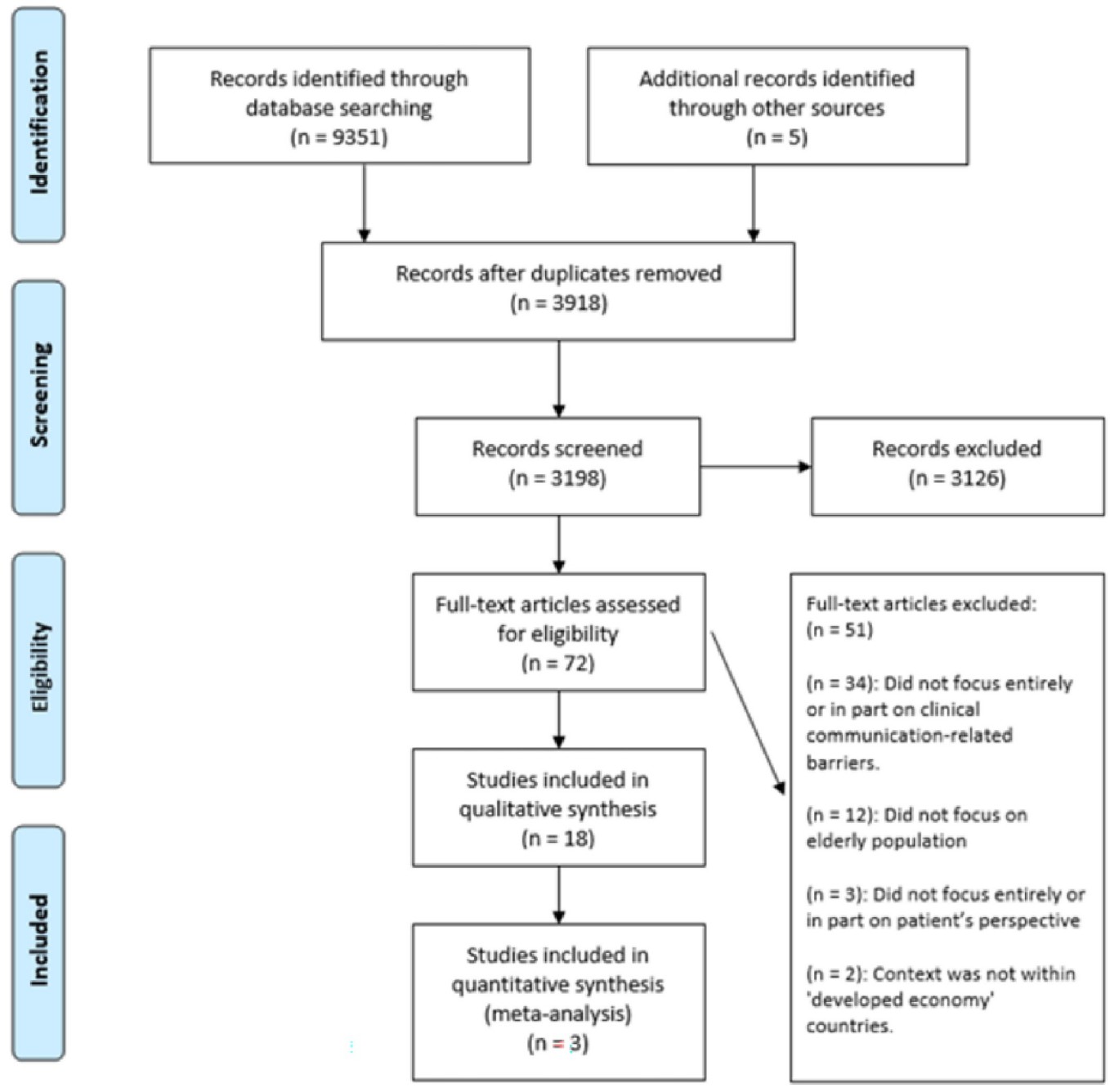

**Fig 1. PRISMA flowchart of the studies retrieved and selection process.**

## Methodological quality

The methodological quality of the 18 qualitative studies scored highly on the CASP scale (Table 1). These high scores also continued throughout the quantitative articles (Table 2). Only minimal methodological deficiencies were found in the 21 studies. The score of the qualitative articles was influenced by the methodological deficiencies described in Table 1. For the quantitative articles, these methodological deficiencies were also identified and highlighted in Table 2.

**Table 3. Characteristics of the qualitative studies.**

| Author(s) and Year | Aim of Study | Population | Method | Main Findings |
|---|---|---|---|---|
| (Aasen et al., 2011) [58] | To explore how elderly patients with end-stage renal disease undergoing treatment with haemodialysis for the rest of their lives perceive patient participation in a dialysis unit. | • n = 11<br>• **Patients:** Haemodialysis patients<br>• Ages: 72–90 | • **Data Collection:** Transcribed interviews<br>• **Location:** Five hospitals in Norway | • Health care teams exercised control and power over the patients. This included withholding information. This power imbalance caused:<br>• Some patients to feel powerless to make their own healthcare decisions–despite trusting the healthcare team.<br>• Patient identity to be threatened over the years of treatment.<br>• Most patients to desire more dialogue about the future.<br>• Barriers to shared decision making about treatment–subsequently threatening patient well-being and quality of life. |
| (Black et al., 2018) [42] | To explore the 'lived experience' of a group of patients with palliative care needs who had recently been in-patients in one acute hospital trust in the north-west of England. | • n = 20<br>• **Patients:** Cancer Patients<br>• Ages: 43–87 | • **Data Collection:** Transcribed interviews<br>• **Location:** UK | Older patients felt that:<br>• Healthcare staff acted compassionately and utilised responsive, patient centred care. This assisted the patients in feeling safe and valued as individuals rather than as a commodity.<br>• These acts of compassion were expressed through small actions such as taking the time to talk and care for the patient while ensuring the patients felt comfortable and treated as a human.<br>• However, over-stretched staff, resources, and modes and manner of communication such as lack of information and poor healthcare provider continuity further negatively impacted patient experience. |
| (Brooks et al., 2016) [62] | To explore the views and experiences of older adults with varying health literacy levels who had attended a falls clinic on their overall experience of the falls clinic, access to the service, and provider–patient interaction. | • n = 9<br>**Patients:** Older adults using a falls clinic<br>Ages: 79+ | • **Data Collection:** semi-structured interviews<br>• **Location:** UK | • Patients felt being recognised as an individual, such as including individual qualities and interests in written communications, was important for building trusting relationships with patients. This positively impacted on the patient's willingness to engage with healthcare professionals and follow their recommendations.<br>• This trust synergised with perceived Healthcare professional credibility, further enhancing patient willingness to participate and follow healthcare professional recommendations.<br>• However, patients felt dissatisfaction with impersonal and overly medicalised written and verbal communications that may not have suited their learning style, preferring clear and simple information delivery. |
| (Butterworth & Campbell, 2014) [63] | To explore older patients' trust in their GPs and their perceptions of shared decision making. | • n = 20<br>**Patients:** From GP surgeries<br>Ages: 65 + | • **Data Collection:** Interviews<br>• **Location:** UK | • Trust was an important influencer for shared decision making.<br>• Patients preferred a GP who they could view as a trusted 'ally'. A GP who could care for patients competently throughout the vulnerable ageing process and their participants' increasing health-information requirement was important.<br>• Factors that negatively impacted patient trust in their GP also impacted patient involvement. Common aspects that influenced trust in the GP were GP communication skills and characteristics, consultation duration, and continuity of care.<br>• Patients expressed trust in GPs who appeared both competent and confident in their abilities. GPs who reaching a mutual understanding regarding utilising patient-centred care facilitated patient involvement. |
| (Choi et al., 2016) [43] | To explore experiences related to hearing loss and barriers to hearing health care among older Korean Americans (KAs) | • n = 19<br>**Patients:** Older Korean Americans<br>Ages: 58+ | • **Data Collection:** Interviews (focus groups)<br>**Location:** US | • Older Korean Americans often had difficulties participating in the medical setting due to poor English-speaking skills which prevented them from communicating their health needs.<br>• Patients stated that their Korean-speaking community GPs were unable to accept any new patients due to being overburdened with the increasing number of older KA's.<br>• Patients felt there was a lack of collaborative communication present even with Korean-speaking physicians. This was perceived to be due to a combination of KA's inclination to endure discomfort or pain and the paternalistic physician-patient relationships held in these clinics.<br>• Most patients held negative opinions about hearing aids. This served as a barrier to initially obtaining hearing care.<br>• Patients felt uncertain about what they needed to do to address their hearing loss symptoms. This was due to a lack of knowledge and health literacy in the hearing health care options available to them. This further served as a clinical communication barrier. |
| (Clarke et al., 2014a) [44] | To gain more insight into how older adults living with chronic pain in the community (and not attending pain clinics) perceive their encounters with healthcare professionals, with a view to informing and improving these interactions. | • n = 23<br>**Patients:** >65 years with self-reported musculoskeletal chronic pain<br>Ages: >65 | • **Data Collection:** Interviews<br>**Location:** Scotland | • Healthcare professionals that were being dismissive rather than supportive and supplying information caused patients to feel anxious. This caused patients to feel their treatments were based on assumptions instead of knowledge of the patients experiences and bodies. |

*(Continued)*

**Table 3.** (*Continued*)

| Author(s) and Year | Aim of Study | Population | Method | Main Findings |
|---|---|---|---|---|
| (Clarke et al., 2014b) [45] | To consider (1) perceived sources of, and explanations for, satisfaction and dissatisfaction with primary care physicians and (2) the strategies that older adults with multiple chronic conditions employ to maximize the care they receive | • n = 35<br>**Patients:** Older adults with multiple morbidities<br>**Ages:** 73 to 91 | • **Data Collection:** Interviews<br>• **Location:** Canada | • Most patients felt they received insufficient care due to actions such as medical consult constraints, ageism, and poor physician qualities.<br>• Many (51%) of patients reported that they felt comfortable with GP's who held certain personal qualities. These included being friendly, open, and trustworthy. Patients felt they could discuss their medical concern with these doctors freely and easily.<br>• However, not all patients felt that their doctors were supportive to all their health care needs. This led to patients feeling uncomfortable when discussing sensitive topics such as sexuality or mental health, as patients felt that their doctors were not willing to adequately address these topics.<br>• Many (43%) patients felt that the complexities of their chronic conditions were not compatible with time constraints of medical consultations. Many spoke about the difficulties in having to make decisions on which acute or chronic health issues to discuss during appointments with GPs.<br>• Some, (26%) of the patients felt that the societal devaluation of older adults (i.e., ageism) was causing barriers to healthcare communication. This was either (1) internalised ageism–where patients viewed themselves as "a drain on the health system" unworthy of physician attention or (2) perspectives of societally generated gendered ageism–where older adult female patients felt male physicians' lack of thoroughness was due to the OAP's female gender.<br>• To combat this, several participants, especially women, utilised several strategies to manage their doctors' impressions of them and maximize the care they received. For example:<br>• Some (28%) used lists during appointments to assist in the prioritization of their multiple health concerns.<br>• Some (11%) utilised companions in their appointments.<br>• Many (34%) utilised other information sources about their conditions. This included the Internet, library books, pharmacists, and other health care professionals.<br>• Some (22%) described assertively asking for specialist referrals or diagnostic tests.<br>• However, most (60%) attempted to manage doctors' impressions of them as genuinely compliant patients. This was done by behaving compliant, "being cooperative" and "doing what they were told". |
| (Costello et al., 2012) [46] | To explore the perceptions of independent-living older adults regarding their physicians' role in promoting physical activity (PA). | • n = 31<br>**Patients:** Independently living older adults (either physically inactive or active)<br>Ages: 60+ | • **Data Collection:** Focus group discussions<br>• **Location:** United States | • Despite raising the topic, patients felt they did not have routine conversations about physical activity with their doctor. Those that did felt that the conversation was not helpful or rewarding. |
| (Dilworth et al., 2012) [47] | To explore the experiences of older people who have been readmitted to hospital following recent discharge to their homes. | • n = 3<br>**Patients:** Older patients who were discharged from a large tertiary referral hospital in NSW Australia and readmitted<br>Ages: 65+ | • **Data Collection:** Interviews<br>• **Location:** Australia | • Participants felt powerless, unheard, disrespected, and left out during their experience of being in hospital. This was mainly due to actions such as:<br>• A lack of information sharing from healthcare professionals to patients.<br>• Patients having their knowledge, values and preferences ignored by healthcare professionals.<br>Receiving mixed messages. |
| (Ellins & Glasby, 2014) [48] | To understand the lived experiences of older people moving across service boundaries by utilising an in-depth narrative approach and adopting a participatory action research method. | • n = 24<br>**Patients:** People who had experienced a recent hospital stay as a patient (or a family member providing care and support)<br>Ages: 60–79 | • **Data Collection:** Interviews (alongside retrospective cohort case studies)<br>Location: UK | Patients felt:<br>• A need for humanistic and person-centred approaches to their care.<br>• There were, at times, difficulties accessing translating services. |
| (Evans et al., 2012) [49] | To examine older patients' attitudes towards, and experiences of, patient-physician end-of-life (EoL) communication in three European countries. | • n = 30<br>**Patients:** British, Dutch and Belgian patients with a terminal illness<br>Ages: 60+ | • **Study Design:** Qualitative<br>• **Data Collection:** Interviews (secondary analysis)<br>• **Location:** UK, Germany, and Belgium | **Patients felt:**<br>• Patients' confidence and trust in healthcare professionals were reinforced by the doctor's time availability, and genuine attention given to the patient.<br>• Poor communication styles from physicians hindered patients' ability to communicate and participate in consults. |
| (Gerlich et al., 2012) [60] | To explore the needs of older patients with advanced heart failure, and their experiences with health care delivery in Germany. | • n = 12<br>**Patients:** Older patients with advanced heart failure<br>Ages: 73 + | • **Study Design:** Qualitative<br>Data Collection: Interviews<br>**Location:** Germany. | • Patients at times played a 'strong role' to avoid speaking about their illness and associated fears.<br>• Patients wished for information to be communicated in an understandable manner. They found it difficult to discover healthcare professionals who meet their information needs. However, patients also stated that too much information may be overwhelming. |

(*Continued*)

**Table 3.** (Continued)

| Author(s) and Year | Aim of Study | Population | Method | Main Findings |
|---|---|---|---|---|
| (Gordon et al., 2018) [50] | To explore older people's accounts of how they talk about depression and possible symptoms to improve communication about depression when seeing GPs. | • **n** = 16 <br> **Patients:** Older patients aged over 65 with depression. <br> • **Ages:** 67–88 | • **Study Design:** Qualitative <br> **Data Collection:** Interviews (secondary analysis) <br> **Location:** North-east England. | Some older patients appeared to: <br> • Deny or minimize their depression due to the perceived stigma and personal insecurities about having depression. This prevented full disclosure in clinical communications. <br> • Due to having depression, believed themselves have greater knowledge about depression than healthcare staff. This subsequently caused anger and distrust of healthcare staff. <br> • Have trouble expressing their feelings and depression and appeared unengaged about working on doing so. <br> • Accept their depression as a part of their personality with no desire to explore their feelings or understand their depression and had no hope about their depression changing or improving. |
| (Pennbrant et al., 2012) [51] | To describe how elderly patients experience their meetings with their doctor in a hospital setting | • n = 20 <br> **Patients:** OAPs discharged from medicine and geriatric hospital care in Sweden <br> **Ages:** 68 to 95 | • **Study Design:** Qualitative <br> **Data Collection:** Interviews <br> **Location:** Sweden | • OAPs better understood their doctors' information with their relatives help. <br> • OAPs were interested in shared decision making and taking an active role in healthcare discussions. However, doctor-patient power relationships often made the patient feel powerless and subordinate to their doctors. <br> • Active interaction improves OAP understanding of their health. OAPs feel that doctors have an obligation to encourage patients active involvement in consultations and show patients that they are the focus of attention. <br> • Older patients are sensitive to the doctors' body language, with negative body language such as appearing rushed reducing patient trust in the doctor. |
| (Schröder et al., 2018) [59] | To analyse socioeconomic differences in patients' experiences along the treatment pathway for coronary heart disease (CHD). | • n = 41 <br> **Patients:** CHD sufferers <br> Ages: 59–80 | • **Study Design:** Qualitative <br> **Data Collection:** Interviews <br> **Location:** Germany | • Socioeconomic status (SES) influences OAP health literacy. OAPs with a lower-SES were less informed about their treatment, which impacted their understanding of health conversations or their health reports in comparison to their higher-SES counterparts. <br> • SES influences OAP participation. Lower-SES patients tended to delegate their responsibility for treatment. This resulted in patients not participating in clinical communications or questioning treatment decisions. In comparison, high SES OAPs felt more responsible for their own treatment and made informed health decisions alongside their GP. |
| (Tobiano et al., 2015)* [52] | To explore hospitalised medical patients' perceptions of participating in nursing care, including barriers and facilitators. | • n = 20 <br> **Patients:** Varied characteristics <br> **Ages:** 18+ (Only data from those >65 was extracted for this study) | • **Study Design:** Qualitative <br> Data Collection: Interviews <br> **Location:** Australia | • Some older patients undertook strategies to gain knowledge from various sources such as nurses, doctors, families, self-education and listening to their bedside handovers from nurses to doctors. This increased health knowledge and enabled patients to become healthcare partners with their nurse. <br> • Some older patients identified that power imbalances consistent with a paternalistic model of health care acted as a barrier to patient participation in clinical communication. These included: <br> • Some OAPs self-perceived that 'nurses held all the power' and that patients participating in their clinical communications would only serve as a burden. <br> • Some OAPs perceived some nurse behaviours as negative, which established a passive role in the patient and prevented patient participation. <br> * Only data that exclusively involved the OAPs (66+) was extracted from this article. |
| (van Ee et al., 2018) [53] | To provide insight into older men's experiences with prostate cancer in order to improve personalised care. | • n = 22 <br> **Patients:** older men with prostate cancer <br> Ages: 70+ | • **Study Design:** Qualitative <br> • **Data Collection:** Interviews <br> • **Location:** Netherlands | • Some OAPs felt appreciative of the use of humour in conversations from healthcare professionals. OAPs looked forward to appointments with these healthcare professionals and reported almost seeing them as outings. <br> • Despite generally positive views of the hospital care, older patients would have preferred to be presented with more information about treatment and support options. |

(*Continued*)

**Table 3.** (Continued)

| Author(s) and Year | Aim of Study | Population | Method | Main Findings |
|---|---|---|---|---|
| (Waterworth et al., 2017) [54] | To determine which aspects of primary nurse–patient telephone communication are viewed positively or negatively in terms of meeting the older persons' needs | • **n** = 21 **Patients:** older patients from General Practices **Ages:** 66–90 | • **Study Design:** Qualitative • **Data Collection:** Interviews (and a focus group) • **Location:** New Zealand | • Accessibility to a healthcare professional was a significant factor in initiating contact with them. Being able to utilise the telephone to access the clinic nurse instead of travelling to the practice in person acted as an enabler to patient contact. • However, potential waiting times for a call back if the nurse was not available was a significant influencer in the decision-making process to initiate contact. Patients often wondered when they would get a call back, or if they ever would. • The financial cost of seeing a GP and its impact on initiating contact was mentioned by several participants. These financial barriers lead to missed opportunities for early intervention. • OAPs were attributing symptoms to ageing and as a result that their symptoms were not urgent or did not matter. Due to this, they felt uncertain about the need to initiate contact with healthcare professionals such as doctors and nurses. |

## Themes of study findings

A total of three themes and three subthemes that addressed the objective were identified across the 21 included articles (Fig 2). The theme '*Accessibility to appointments, support, health information and person-centred care*' explores how patient access to the three subthemes; (1) appointments, (2) person centred care, and (3) support and health information resources influence OAP participation. The theme '*relevant and understandable healthcare information*' explores how communication factors influences OAP participation. The theme '*Older patient perceptions of HCP credibility and trustworthiness*' explores how patient's perception of healthcare professional credibility and trustworthiness influences their participation.

## First theme: Accessibility to appointments, support, health information and person-centred care

The central theme of how OAP participation is influenced by accessibility to factors such as: (i) appointments, (ii) person centred care, and (iii) resources were discussed in articles to a significant extent. Of the 21 articles included, 18 (16 qualitative and 2 quantitative) elucidated the influence that accessibility to these factors had on OAP participation in health-related clinical communications [13,42–56].

**Accessibility to appointments.** Some (5) studies have shown that there are several HCP and patient-generated factors that influence OAPs' perception of appointment access and, consequently, attendance at clinical appointments. HCP-generated factors generally encompassed overstretched health services. For example, one study highlighted that elderly Korean-speaking patients in the US had difficulty reaching Korean-speaking GPs because they were not accepting new patients due to their overload [43]. Another study found that older patients with mood disorders had greater difficulty accessing care, especially after hours, compared to patients without mood disorders [55]. Another study found that older adults were reluctant to initiate and participate in telehealth consults with nurses due to long and sometimes unknown wait times for callbacks [57]. However, the same study also concluded that being able to use telephone appointments to reach the nurse in the clinic, rather than coming to the practice in person, allowed for better patient contact [54]. A common theme that impacted OAPs engaging in appointment making were perceptions that symptoms are an inevitable consequence of aging. For example, two studies identified that the decision to seek help was influenced by the belief that "nothing much" could be done about pain, as they associated it with getting older [44,54]. As a result, participants did not think their symptoms were urgent or important [54].

**Table 4. Characteristics of the quantitative studies.**

| Author(s) and Year | Aim of Study | Population | Method | Main Findings |
|---|---|---|---|---|
| (Fortuna et al., 2017) [55] | To compare patient experience with healthcare services and providers among older patients (≥50 years old) with and without serious mental illness. | • **n** = 35446 **Patients:** Schizophrenia (n = 106)mood disorders (n = 419)no serious mental illness (n = 34,921)**Ages:** 50+ | • **Study Design:** Quantitative • **Data Collection:** Secondary data from the Medical Expenditures Panel Survey from 2003 to 2013 • **Location:** US | • Older Patients with mood disorders found it more difficult to access healthcare compared to those without Serious Mental Illness. This was especially the case when trying to contact their healthcare provider after hours. |
| (Foss & Hofoss, 2011) [61] | To describe older hospital patients' discharge experiences specifically related to participation in the discharge planning. | • **n** = 254 • **Patients:** first 2–3 weeks (mean 19.2 days) following discharge from hospital **Ages:** 80 + | • **Study Design:** Quantitative • **Data Collection:** Interviews • **Location:** Oslo, Norway | • Poor hearing ability was a factor that negatively affected the patient's capacity to participate in clinical settings. |
| (Gibney & Moore, 2018) [56] | To investigate the link between provider communication and older patients' perceived encouragement to talk about physical, social, sensitive, and emotional problems with their usual source of care (USC), be it a doctor or nurse | • **n** = 720 • **Patients**: Older patients from Ireland • **Ages**: 50+ | • **Study Design**: Quantitative • **Data Collection:** Irish sample of the Survey of Health, Ageing and Retirement in Europe • **Location:** Ireland | • The nature of the health problem influenced patient participation. More patients felt discouraged to talk about health problems that were sensitive or their social issues when compared to health problems that were physical or emotional in nature. |

They also felt a reluctance to see their GP and "bother" them, "waste" their time or be seen as "complainers" by visiting GPs "too often" [44,54].

**Accessibility to person centred care.** Factors that impact accessibility to person centred care-focused HCP's and how this affects OAP participation was discussed in most studies (13). Several studies discussed factors that impacted person-centred care communication and thus patient participation, including collaborative communication (6), power imbalances (6), time to listen (3), structure of doctor's appointments (1), and the issue of clinical communication (1).

OAPs had difficulty engaging in clinical communication with medical staff who lacked openness, support, responsiveness or collaborative communication about treatment options, or who seemed preoccupied or unavailable [42,43,45]. The following quote illustrates how the perceived availability of health workers can affect patient participation: *"...sometimes they were run off their feet. They can't always come, so you don't get bad tempered or anything, you just have to wait and know that they will come."* [42]. One study revealed that one factor affecting HCP collaborative approaches was the presence of a mood disorder in patients. Elderly patients with a mood disorder were more likely to report that their doctors did not ask them to help make decisions regarding choice of treatments compared to those without a mood disorder [55]. Conversely, one study found that many older patients experienced collaborative, joint discussions with their HCP about treatment or non-treatment options [49].

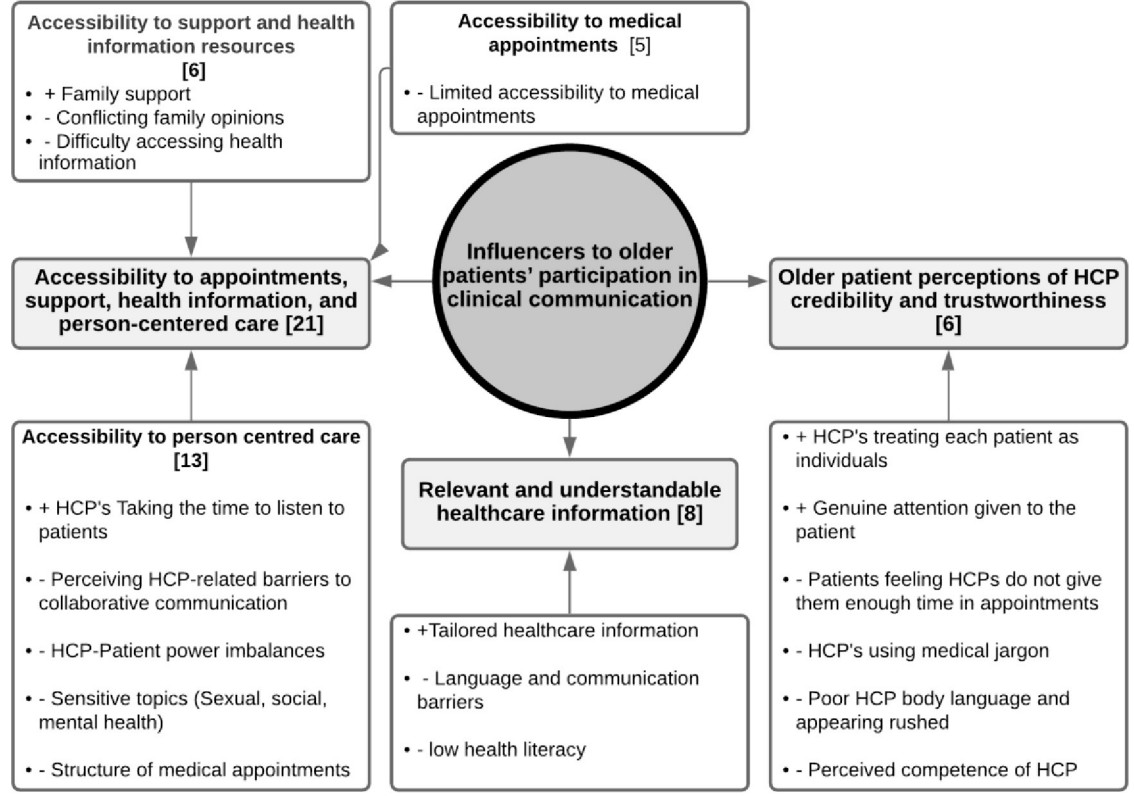

**Fig 2. Flowchart highlighting number of studies that discussed each topic theme and sub-theme, and the influence (positive or negative) of each theme on older patient participation in clinical communication settings.**

The perceived or actual HCP-patient power imbalances were also a significant factor influencing OAP participation in clinical communication. Several studies found that despite patients' interest in participating in their treatment, asymmetrical power relations between doctor-patient power relationships often made them feel powerless, leading to passivity in clinical communication [43,52,58]. This was illustrated by the quotes in the Clarke et al. study, where many (23%) of the OAPs stated that it was important to "do as you're told" by their physicians, and 40% emphasised that it was important to "not waste the doctor's time [45]". Some studies attributed these power imbalances to HCP-generated factors, namely communication-related, while other studies uncovered patient-generated factors. For example, Schröder et al. found that lower-SES OAPs tended to delegate their responsibility for treatment to HCP's [59].

HCP's taking the time to listen to OAPs was another significant factor influencing patient centred care and, subsequently, patient participation. Some studies identified the positive impact listening can have on patient participation. OAPs felt "fortunate" to have HCP's who took the time to listen to them and felt more inclined to participate in clinical communication [42,44]. Furthermore, Dilworth et al. illustrated how a lack of listening can severely affect patients' satisfaction and ability to participate with the following quote: *"They haven't told me ANYTHING was the matter with me as far as I know. . .It's all I want to know what is the matter with me?"* [47].

There were also less common, but still influential, person-centred communication accessibility issues that influenced OAP participation. One factor was sensitive communication issues such as social problems, sexuality or mental health problems [45,56]. Another factor was the

typical structure of medical appointments, which was perceived as incompatible with discussing the complex chronic conditions associated with OAPs [45]. Finally, some OAPs wanted to avoid communication of their illness and subsequently, played a 'strong role' to do so [60].

**Accessibility to support and health information resources.** OAPs' accessibility to support and health information resources was also an important factor influencing their participation in clinical communication. Patients felt that family support gave them a sense of security and improved their clinical understanding, enabling them to take an active role in clinical communication [49,51]. This is reflected in the quote *"It feels good that my daughter works in health care. . . She is a nurse so if I wonder about something I can ask her how it should be"*. However, some patients felt that family support can have a negative impact on patients' participation in healthcare due to conflicting expectations about healthcare and treatment [49].

Accessibility of health information was also a factor influencing health literacy and patient participation in clinical communication. Diagnoses seemed to motivate patients to participate because they had an increased need for information and the information was obtained by the patients HCP through clinical communication [53]. However, difficulties in accessing health information from HCPs had a significant negative impact on patient participation. In some studies, patients reported that their ability to participate in clinical communication was affected by difficulties in accessing health information from their HCP [46,48]. Lack of access to health resources contributed to low health literacy and uncertainty about navigating healthcare and acted as a barrier to patient participation [43]. For example, unreliable resources affected patient participation because of misconceptions about the treatability of conditions such as depression or hearing loss [43,57]. This concept is illustrated by the following quote: *"I saw a TV program that talked about hearing once. . .The conclusion was that there is no solution for hearing loss."* [43].

## Second theme: Relevant and understandable healthcare information

Another predominant theme identified in the analysis was the positive impact that relevant and understandable clinical communication has on patient communication. Overall, older patients who felt better informed about their disease were able to share their understanding and participate in healthcare communication [52]. Consequently, factors that influence patient understanding, such as the relevance and comprehensibility of HCP communications, may subsequently influence OAP participation. Factors influencing the relevance and comprehensibility of healthcare were elicited from both HCP and patients. Of the 21 articles included, 8 (7 qualitative and 1 quantitative) addressed this issue [43,48,49,52,57,59–61]. There were several HCP-driven factors that affected how relevant and understandable healthcare communication was to OAPs, which subsequently influenced their ability to participate. For example, three studies found that patients preferred to receive health information tailored to their individual needs and HCP's that tried to explain medical terms that were difficult to understand [49,60,62]. OAPs were dissatisfied with impersonal and overly medicalised written and verbal communications and preferred clear and simple information delivery [62]. For example, one participant in the Brooks et al. study described a standardised sheet of printed information that was not relevant to anyone's situation because it did not take into account other health conditions [62].

Patient-related factors focused on health literacy, language barriers and communication impairments, including language and dialect comprehension and disabilities such as strokes and hearing impairments [43,48,61]. One study found that dialect-related barriers to participation were exacerbated by difficulties in accessing interpreter services during hospitalisation [48]. Although hospitals offered interpreting services, these were not always adequately

available when needed [48]. In one case, the presence of bilingual cleaners in the facility was the reason why a professional interpreter was not requested for a patient [48]. Some patients, particularly those of low socioeconomic status, had low health literacy about their condition, which affected their ability to understand health conversations [59]. This is illustrated by the following quote: *"And as well, someone who is cognitively not that fit anymore, one does not understand at all what you are told. And that is sometimes not so nice. Well, but probably that's just our system, that is/which is not working, I think"* [59]. However, some patients combated this by adopting strategies to acquire knowledge from various sources such as nurses, doctors, family members, self-education and listening during bedside handovers [52].

### Third theme: Older patient perceptions of HCP credibility and trustworthiness

The final main theme identified related to the factors that determine the OAP's perceptions of health professionals' credibility and trustworthiness and subsequently influence their participation. Overall, trusting HCP-OAP relationships motivate OAPs to participate in shared decision-making and to be receptive to messages from health professionals [62–64]. Of the 21 included articles, 6 articles (all qualitative) explored this topic [45,49–51,62,63].

Several factors influenced perceived HCP credibility and trustworthiness and, subsequently, patient engagement in clinical communication. Communicative factors were frequently mentioned, where OAPs expressed they lost confidence and trust in healthcare professionals who used medical jargon [49]. Conversely, OAPs expressed that they were able to communicate freely and easily with HCPs who were caring, attentive, friendly, open and recognised them as individuals [45,62,63]. This phenomenon is illustrated by the following quote: *"I trust him so much that any problem I had I would tell him. He talks to me very plainly about my health issues and what I can do. He gives me all the options I have. . . and then we discuss it."* [45].

Patients' perceptions and the time factor also played a major role. Patients' trust in health professionals was enhanced by the availability of time and the genuine attention they gave to the patient [49]. Butterworth & Campbell found that time seemed to become more important with age and patients felt less trust and commitment when they did not have enough time at appointments [63]. This was corroborated by Pennbrant et al. who found that older patients are sensitive to the doctor's body language and that negative body language, such as a rushed demeanour, reduces patients' trust in the doctor [51]. This is reflected in the quote: *"In a rush, no time to inform, too much to do. . . Not enough doctors, they are burnt out and can't cope. . . No worthwhile information, they had so much to do. . . His body language showed he was in a hurry"* [51].

In terms of patient perceptions, the patient's assessment of HCP competence also influenced their assessment of HCP trustworthiness and credibility, subsequently influencing patient engagement [63]. This factor was evident in the study by Gordon et al. in which patients believed they knew more about depression than health care practitioners because of their depression—leading to mistrust of them [50].

### Discussion

The aim of this systematic review was to capture the literature that addresses the factors that contribute to older patients engaging with HCPs in hospitals and GP clinics. In the 21 studies in this review, the importance of accessibility to appointments, person-centred care, support and health information sources was a common theme. This suggests that older patients consider these factors important for their participation in clinical communication. These findings also correlate with the literature across all age groups of patients. For example, younger

patients (< 17) value accessibility but are best influenced by a person-centred, safe and comfortable atmosphere [68,69]. Furthermore, people of different age groups valued different modes of accessibility for their participation. Young adults (19–39) preferred mobile technologies and text messaging, middle-aged adults (40–64) preferred phone calls, and older adults (> 65) preferred direct HCP interactions [70]. This suggests that OAP participation could be positively influenced by a HCP-patient interaction framework that prioritises accessibility to person-centred, face-to-face communication.

Many of the main models of HCP-Patient communication lack the capacity to empower patients to participate in clinical communication [57]. This research has demonstrated the importance of perceived and actual accessibility of health services in promoting older patients' participation in clinical communication. Ringdal et al. found that patients felt safe and encouraged to communicate openly when the interaction was person-centred and the healthcare team was accepting [65]. This highlights that accessibility to person-centred care significantly impacts OAP participation, as highlighted in this systematic review.

This supports the idea that person-centred care has a positive impact on patient participation and reduces the number of 'missed' appointments in the clinic. Similar to the review, several studies highlighted that (1) social support and (2) health information sharing positively influence patients' health literacy, self-perceived safety and participation in clinical communication [65–67,71,72]. This ultimately underlines the importance of access to support and health information for improving OAP health literacy and participation in clinical communication.

Some of the main paternalistic models of HCP-Patient communication may result in insufficient clinical communication and patient understanding, particularly among those with low health literacy [28]. Another theme focusing on health literacy was uncovered in relation to 'relevant and understandable healthcare information'. This theme explored the communication techniques used by HCP and the patient ability to understand information, and the impact of these factors on health literacy and subsequently on patient engagement and 'missed care'. The finding that older patients prefer and respond better to tailored health information compared to impersonal or complicated messages is consistent with previous studies. For example, Santana et al. found that positive HCP communication skills better enable patients to actively participate in their own care, which ultimately leads to better patient outcomes [7]. The concept that language barriers, communication difficulties and impaired hearing affect patient participation is also consistent with the literature. For example, Stransky et al. found that patients with communication disabilities have more difficulty participating in health communication than patients without communication disabilities [73].

Some models of HCP-Patient communication may not take into account common perceptions and misunderstandings in OAPs. A common communication barrier identified throughout the study was patients' misconceptions that their symptoms were an inevitable consequence of ageing and not a sign of disease. This was also a common theme in the wider literature on other age groups. For example, Sun & Smith found that people with poorer self-perceptions of ageing are more likely to delay seeking medical care, which may exacerbate the phenomenon of 'inadequate care' [74]. This is a notable misconception that should be addressed in rapidly ageing populations to avoid further communication barriers.

Several models of HCP-Patient communication may not consider the perceptions OAPs surrounding factors such as appointment availability. The perceived or actual availability of appointments affected patients' participation in clinical communication—particularly in relation to the first appointment. This phenomenon was unexpectedly unique and, to our knowledge, does not appear in any existing literature dealing with developed economies. However, it does appear in the literature on non-developed countries. For example, Gordon et al. pointed

out that people in South Africa who had access to healthcare because of financial means had a greater desire to use it than their economically disadvantaged counterparts [75]. There may be several reasons for these unexpected findings. The first is related to the fact that older people in industrialised countries consider illness a natural part of aging, and do not feel the need to visit HCP's. In addition, most of the studies in this review were conducted in large cities, where accessibility is less of a problem than in rural and remote communities. This indicates a gap in the literature that could be addressed in future studies.

One aspect of the perception of OAPs that may not be addressed by existing models of HCP-Patient communication is the perception of HCP credibility and trustworthiness. Patients' perceived HCP credibility and trustworthiness largely influenced older patients' participation in clinical communication. The finding that older patients were more likely to participate if they perceived their HCP as credible and trustworthy was an expected result, consistent with previous studies. For example, Leslie & Lonneman have highlighted that a trusting relationship between nurse (RN)—patient promotes patient participation and increases the chances that the patient will be an active member of the healthcare team [76]. The fact that time availability, genuine attention and effective communication are important factors in patients' perceived HCP credibility and trust in the nurse is also consistent with the studies of Chichirez & Purcărea and Leslie & Lonneman [76,77].

## Strengths and limitations

This review was conducted using an extensive literature search and rigorous systematic methods. To improve the scope and depth of this review, both qualitative and quantitative studies were included. In addition, strict inclusion and exclusion criteria were applied to improve the specificity of this review and best achieve the intended objectives. An acceptable number of articles (21) were eligible for inclusion in this review. To further increase the richness of data and breadth of perspectives, this review focused on interactions between patients and HCPs from different disciplines.

Methodological quality was rigorously assessed using high-quality appraisal tools, including the Critical Appraisal Skills Programme (CASP) and the JBI Critical Appraisal Checklist for Analytical Cross-Sectional Studies. This allowed for a consistent and systematic assessment of the studies. The overall methodological quality of the studies was moderate to high, with most being high quality.

This review focused on the question of what factors influence participation in clinical communication in the older adult population, regardless of socioeconomic status and geographic location, provided it is a WESP country with a developed economy. We originally intended to focus on rural and remote areas for people aged 65 and over in this study. However, only a limited number of studies were identified that focused both on rural areas and exclusively on people aged 65 and over. Therefore, most focused on large cities and people over 50, resulting in a representative population that was geographically heterogeneous and included a wider range of ages. Therefore, the results need to be interpreted with caution—especially with regard to the results on accessibility to appointments, as accessibility to healthcare is different in rural and remote areas than in metropolitan areas [78].

Despite these limitations, this study provided valuable insights into the factors that influence participation in clinical communication from the OAP perspective. This helped to identify the elements that OAPs considered most important in influencing their participation in clinical communication. The results of this analysis may help to identify ways to improve OAPs' participation in clinical communication in the future.

## Conclusions and implications

This review has highlighted the factors that stand in the way of older patients' participation in clinical communication, often leading to inadequate or inappropriate treatment. Older patients' accessibility to appointments, person-centred care, support and health information resources has a significant impact on participation in clinical communication. In addition, it is of great importance that communication is relevant and understandable to OAPs and that HCPs are perceived as credible and trustworthy. Overall, older patients were more likely to engage in person-centred, accessible clinical communication that was delivered in a relevant and understandable way by HCPs who were perceived as credible and trustworthy. Conversely, older patients reacted negatively and were less likely to engage in inaccessible health-care that was communicated impersonally or convolutedly by HCP's that were perceived as unknowledgeable or untrustworthy. Uncovering these positive and negative factors can be beneficial for both health practitioners and older patients when it comes to health-related communication. These themes can be used to aid HCP's in designing person-centred questions that will aid their older patients to ask questions about their diseases, treatment options and medications. This in turn can improve patient empowerment, patient participation, doctor-patient collaboration and overall health outcomes for OAPs. The limited number of studies included in the review that include participants from rural areas reflects the knowledge gap for this population. To address this research gap, additional research on factors influencing OAP participation in clinical communication in rural and remote areas is recommended.

## Supporting information

**S1 Checklist.**
(PDF)

**S1 Protocol. A link to the systematic review protocol located in the PROSPERO International prospective register of systematic reviews.** https://www.crd.york.ac.uk/PROSPEROFILES/164716_PROTOCOL_20220124.pdf.
(PDF)

**S1 Text. An example of a search string applied to the CINAHL database to elucidate relevant articles.**
(DOCX)

## Author Contributions

**Conceptualization:** Harry James Gaffney, Mohammad Hamiduzzaman.

**Data curation:** Harry James Gaffney.

**Formal analysis:** Harry James Gaffney.

**Funding acquisition:** Harry James Gaffney.

**Investigation:** Harry James Gaffney.

**Methodology:** Harry James Gaffney.

**Project administration:** Harry James Gaffney.

**Resources:** Harry James Gaffney.

**Software:** Harry James Gaffney.

**Supervision:** Mohammad Hamiduzzaman.

**Validation:** Harry James Gaffney.

**Visualization:** Harry James Gaffney.

**Writing – original draft:** Harry James Gaffney.

**Writing – review & editing:** Harry James Gaffney, Mohammad Hamiduzzaman.

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
