## [Decision Letter · Decision Letter 0]

11 Jan 2022

PONE-D-21-29170Influencers to older patients’ participation in clinical communication within hospitals and GP Clinics: A systematic review of current literature.PLOS ONE

Dear Dr. Gaffney

Thank you for submitting your manuscript to PLOS ONE. After careful consideration, we feel that it has merit but does not fully meet PLOS ONE’s publication criteria as it currently stands. Therefore, we invite you to submit a revised version of the manuscript that addresses the points raised during the review process.

We look forward to receiving your revised manuscript.

Kind regards,

Edris Hasanpoor

Academic Editor

PLOS ONE

2. Please report your search date in the Method section.

“The funders had no role in study design, data collection and analysis, decision to publish, or preparation of the manuscript”

Reviewers Comments:

Reviewer comments 1:

The study dealt with an important topic of great interest in medicine, as it can aid in providing high-quality medical care to the elderly. However, I will mention some comments and notes I found during my review of manuscript. I think they should be addressed so the submission will be compatible with the criteria of publication. - The title of the study must specify that this systematic review includes developed countries only. - In the field of Article type, the type of study has been determined as research article but it is better to specify as a (Systematic review) - The abstract was not written in the manner usually used in PLOS ONE. In addition, it did not mention the objective of the study or anything about the results. Also, the conclusions were not written correctly (what is written in it is closer to be recommendations or rationale. There is a need to provide more information about the research method. - It must be clarified whether there is funding for the research or not. At the beginning of manuscript, it was written that the funders had no role in study design, data collection and analysis, decision to publish, or preparation of the manuscript, while later it was written that there is no funding. - There is some lengthening in the introduction part and repetition of some topics. For example, factors affecting HCP-patient communication (from patient side) were mentioned in two times, first in lines 65-66 and then repeating them again in lines 84-85. On the other side, other related topics to study subject were left out, such as the effect of good communication on patient satisfaction and treatment outcome, which was just mentioned in brief in lines 66-67 without any detail or explanation. - It was a good thing to write the protocol, guidelines operators and terms used in the review, even it need to be clarified a little more (at least in an appendix to the study or as supplementary file). - The section of search strategy in method part is better to be divided into several sections (e.g. Protocol and registration, Search strategy, Inclusion/ eligibility criteria). Additionally, it will be better if the literature search strategy was explained in a flow diagram or chart. - The following issues were not addressed in method section: 1. the date of beginning of this systematic; review and the time period it took, 2. the exclusion criteria, and 3. risk of bias assessment - It is better to present the findings of Critical Appraisal of reviewed studies in the form of a short tables, while the details are clarified in an appendix/ supplementary file added to the study. - The Results of systematic review was never made clear in this manuscript. The section of Themes of study findings in line 27 (which represents the results as I think) has not explained the findings in a clear way despite the lengthening and the frequent use of quotes that did not add much. It was more useful for authors to use some tables and figures in this section to indicate the number of studies that discussed the topic themes and sub-themes, as well as to indicate the most important common shared finding among them. This method will facilitate the understanding for the persons who will read the article. - In the discussion section there are repetitions of some phrases, for example: the phrase "many major models of HCP-Patient communication may also not consider the perception…" has been mentioned almost with the same meaning more than once. In addition, the sentences in the lines 535-536 "Another older adult patients’ perception Many major models of HCP-Patient communication may also not consider are perceptions of HCP credibility and trustworthiness." are not clear or connected to me - The acknowledgment (line 597-598) should not be from one author to another author who contributed to the study. - The submission requires copyediting for English usage and grammar because it contains many linguistic mistakes and grammatical errors in formulating phrases and sentences..

Reviewer comments 2:

hank you for giving me the opportunity to review this paper. Please find my comments below: Introduction • References are needed for these sentences: 1. “Increasing disease burden at old age is associated with complex care needs and frequent use of GPs, hospitals, and emergency departments. Without adequate participation of older patients in clinical communication, health care and support may remain inappropriate and ineffective for them”. 2. “patient-driven factors such as attitudes, perceptions, health literacy, and participation are equally relevant to effective clinical communication”. Materials and Methods • Please provide clarification and what are the abbreviations for (PICO framework and PRISMA) stand for. • I recommend moving the search string applied to the CINAHL database (the second paragraph) into the appendices if that conforms with the journal guidelines. • Inclusion criteria for this review were full-text, peer reviewed, qualitative or quantitative. I suggest including these criteria in numbering, i. full text ii. Peer review iii. Qualitative or quantitative ……………… • Inclusion criteria: carried out in comparable countries to Australia defined. Could you please provide a reason for using this an inclusion criterion? • I suggest including exclusion criteria in the text (similar to those mentioned in the PRISMA flow diagram). • I suggest to change the heading of data collection into identification of studies. • I recommend also to move the tables of critical appraisal into the appendices. Also I recommend to provide some background about the tools used in quality assessment in the text in the methods section. • I suggest including section about data synthesis/analysis including the details of thematic analysis after the data extraction section. Results • I suggest making methodological quality more concise. • I suggest renaming the identified themes to reflect the study objectives. • Line 422- Some patients, particularly those with low socioeconomic status, held low health literacy about their condition and this impacted their capacity to 422 understand health conversations – impacting their participation (Schröder et al., 2018). Please fix the reference style to be consistent throughout the paper. Discussion • Please fix this sentence. Line 535-Another older adult patients’ perception Many major models of HCP-Patient communication may also not consider are perceptions of HCP credibility and trustworthiness. Conclusions • I recommend changing the heading into conclusions and implications.

---

## [Author Response · Author response to Decision Letter 0]

24 Jan 2022

***Responses to reviewer 1***

The study dealt with an important topic of great interest in medicine, as it can aid in providing high-quality medical care to the elderly. However, I will mention some comments and notes I found during my review of manuscript. I think they should be addressed so the submission will be compatible with the criteria of publication. 

1. The title of the study must specify that this systematic review includes developed countries only. - In the field of Article type, the type of study has been determined as research article but it is better to specify as a (Systematic review) 

• Changed title to “Influencers to older patients’ participation in clinical communication within developed country hospitals and GP Clinics: A systematic review of current literature.” (Lines 4-6)

2. The abstract was not written in the manner usually used in PLOS ONE. In addition, it did not mention the objective of the study or anything about the results. Also, the conclusions were not written correctly (what is written in it is closer to be recommendations or rationale. There is a need to provide more information about the research method. 

• Objective has been added into abstract (Lines 25-28)

• The research method has been adjusted for clarity and have more information added (Lines 29-31)

• The results have been adjusted to ensure clarity (Lines 32-41)

• The conclusions of the review has been adjusted for clarity (Lines 42-47)

3. It must be clarified whether there is funding for the research or not. At the beginning of manuscript, it was written that the funders had no role in study design, data collection and analysis, decision to publish, or preparation of the manuscript, while later it was written that there is no funding. 

• The authors received no specific funding for this work. This statement has been added into the cover letter. It has also been added to the manuscript (Lines 532-533)

4. There is some lengthening in the introduction part and repetition of some topics. For example, factors affecting HCP-patient communication (from patient side) were mentioned in two times, first in lines 65-66 and then repeating them again in lines 84-85. On the other side, other related topics to study subject were left out, such as the effect of good communication on patient satisfaction and treatment outcome, which was just mentioned in brief in lines 66-67 without any detail or explanation. 

• The introduction has been significantly shortened and additional justification for how communication influences patient satisfaction and treatment outcome (via its influence in missed care) has been added. (Lines 50-102)

5. It was a good thing to write the protocol, guidelines operators and terms used in the review, even it need to be clarified a little more (at least in an appendix to the study or as supplementary file).

• The protocol has been added as a supplementary file to the resubmission (Page 43, under S1 Protocol).

6. The section of search strategy in method part is better to be divided into several sections (e.g. Protocol and registration, Search strategy, Inclusion/ eligibility criteria). Additionally, it will be better if the literature search strategy was explained in a flow diagram or chart. 

• The section of search strategy in the method part has been divided into several sub-sections including Protocol and registration, Search strategy, Inclusion/ eligibility criteria (lines 104-168)

• The search strategy has been explained as flow diagram via the PRISMA protocol (Fig 1, page 41)

6. The following issues were not addressed in method section: 

1. the date of beginning of this systematic; review and the time period it took, 

2. the exclusion criteria, and 

3. risk of bias assessment - It is better to present the findings of Critical Appraisal of reviewed studies in the form of a short tables, while the details are clarified in an appendix/ supplementary file added to the study. 

• The search beginning and end dates has been added into the Method section (lines 106-107) 

• The exclusion criteria has been added (lines 137, 140)

The critical appraisal processes have been tabulated (Tables 1 and 2, Pages 7-9)

7. The Results of systematic review was never made clear in this manuscript. The section of Themes of study findings in line 27 (which represents the results as I think) has not explained the findings in a clear way despite the lengthening and the frequent use of quotes that did not add much. It was more useful for authors to use some tables and figures in this section to indicate the number of studies that discussed the topic themes and sub-themes, as well as to indicate the most important common shared finding among them. This method will facilitate the understanding for the persons who will read the article. 

• The results have been clarified by adding in a relevant figure which highlights how the results report on the outcomes to the articles objective (i.e. Influencers to older patients’ participation in clinical communication within developed country hospitals and GP Clinics) (page 42)

• The results have been restructured for clarity and superfluous quotes have been removed. (Lines 194-396)

8. In the discussion section there are repetitions of some phrases, for example: the phrase "many major models of HCP-Patient communication may also not consider the perception…" has been mentioned almost with the same meaning more than once. 

• The discussion section has been altered to remove repetitive phrases and increase overall clarity 403-476

9. In addition, the sentences in the lines 535-536 "Another older adult patients’ perception Many major models of HCP-Patient communication may also not consider are perceptions of HCP credibility and trustworthiness." are not clear or connected to me 

• This sentence has been addressed during the copyediting process. 

10. The acknowledgment (line 597-598) should not be from one author to another author who contributed to the study. 

• This acknowledgement has been removed. 

11. The submission requires copyediting for English usage and grammar because it contains many linguistic mistakes and grammatical errors in formulating phrases and sentences.

12. The manuscript has been copyedited for English usage and grammar, and all linguistic mistakes and grammatical errors in formulating phrases and sentences have been addressed. 

***Responses to reviewer 2***

Thank you for giving me the opportunity to review this paper. Please find my comments below: 

Introduction

References are needed for these sentences: 

1. “Increasing disease burden at old age is associated with complex care needs and frequent use of GPs, hospitals, and emergency departments. Without adequate participation of older patients in clinical communication, health care and support may remain inappropriate and ineffective for them”. 

2. “patient-driven factors such as attitudes, perceptions, health literacy, and participation are equally relevant to effective clinical communication”. 

• References have either been provided or the sentences have been removed during the copyediting process. 

Materials and Methods

Please provide clarification and what are the abbreviations for (PICO framework and PRISMA) stand for. 

• Acronyms (PICO PRISMA) have been clarified (Lines 108-110)

I recommend moving the search string applied to the CINAHL database (the second paragraph) into the appendices if that conforms with the journal guidelines. 

• The search string applied to the CINAHL database has been moved into the appendices (S2 Text)

• Inclusion criteria for this review were full-text, peer reviewed, qualitative or quantitative. I suggest including these criteria in numbering, i. full text ii. Peer review iii. Qualitative or quantitative ……………… 

• The inclusion criteria have now been laid out this way (Lines 119-126)

Inclusion criteria: carried out in comparable countries to Australia defined. Could you please provide a reason for using this an inclusion criterion? 

• The reason for using developed economy coutries has now been addressed (The decision to include only ‘developed economy’ countries was due to the lack of literature on this subject originating from countries classified otherwise). (Lines 126-128)

I suggest including exclusion criteria in the text (similar to those mentioned in the PRISMA flow diagram). 

• The exclusion criteria has now been included in the text (lines 138-140)

I suggest to change the heading of data collection into identification of studies. 

• The heading data collection has been changed to identification of studies (Line 143)

I recommend also to move the tables of critical appraisal into the appendices. 

• PLOS guidelines require tables to be inserted into the manuscript directly after they are mentioned. We are unable to move these tables into the apendix

Also I recommend to provide some background about the tools used in quality assessment in the text in the methods section. 

• Added some background about the tools used in quality assessment in the text in the methods section (lines 158-167)

I suggest including section about data synthesis/analysis including the details of thematic analysis after the data extraction section. 

• This section includes details of thematic analysis process after data extraction. As a result, the section title was changed to ‘Data extraction, synthesis, and analysis (Line 176)

Results

I suggest making methodological quality more concise 

• The methodological quality section has now been made more concise (lines 215-221).

I suggest renaming the identified themes to reflect the study objectives. 

• Identified themes have been renamed to better reflect the study objectives and their findings (Lines 233-367)

• Line 422- Some patients, particularly those with low socioeconomic status, held low health literacy about their condition and this impacted their capacity to 422 understand health conversations – impacting their participation (Schröder et al., 2018). Please fix the reference style to be consistent throughout the paper. 

• This sentence has now been fixed (Lines 358-360)

Discussion

• Please fix this sentence. Line 535-Another older adult patients’ perception Many major models of HCP-Patient communication may also not consider are perceptions of HCP credibility and trustworthiness.

• This sentence has been fixed during the copyediting process. 

Conclusions

I recommend changing the heading into conclusions and implications.

• The heading has been changed into conclusions and implications (line 508)

---

## [Decision Letter · Decision Letter 1]

14 Mar 2022

PONE-D-21-29170R1Factors that influence older patients’ participation in clinical communication within developed country hospitals and GP Clinics: A systematic review of current literature.PLOS ONE

Dear Dr. Gaffney

Thank you for submitting your manuscript to PLOS ONE. After careful consideration, we feel that it has merit but does not fully meet PLOS ONE’s publication criteria as it currently stands. Therefore, we invite you to submit a revised version of the manuscript that addresses the points raised during the review process.

We look forward to receiving your revised manuscript.

Kind regards,

Edris Hasanpoor

Academic Editor

PLOS ONE

Reviewers' comments:

The authors have responded to the vast majority of the observations I have raised in a manner that enriches the research, but I believe that there is still a need for a slight additional revision of the grammatical and language used and the punctuation between the phrases and in the sentences used in the manuscript.Also, you can use these references:Bahadori, M., Hasanpoor, E., Yaghoubi, M. and HaghGoshyie, E., 2020. Determinants of a high-quality consultation in medical communications: a systematic review of qualitative and quantitative evidence. *International Journal of Human Rights in Healthcare*.Hajebrahimi S, Janati A, Arab-Zozani M, Sokhanvar M, Haghgoshayie E, Siraneh Y, Bahadori M, Hasanpoor E. Medical visit time and predictors in health facilities: a mega systematic review and meta-analysis. International Journal of Human Rights in Healthcare. 2019 Nov 28.Bahadori M, Yaghoubi M, Haghgoshyie E, Ghasemi M, Hasanpoor E. Patients’ and physicians’ perspectives and experiences on the quality of medical consultations: a qualitative study. JBI Evidence Implementation. 2020 Jun 1;18(2):247-55.Hasanpoor E, Bahadori M, Yaghoubi M, Haghgoshayie E, Mahboub-Ahari A. Evidence-based management as a basis for evidence-based medical consultation. BMJ evidence-based medicine. 2020 Jun 1;25(3):83-4..

---

## [Author Response · Author response to Decision Letter 1]

18 Mar 2022

Responses to Journal requirements

Reviewers' comments:

The authors have responded to the vast majority of the observations I have raised in a manner that enriches the research, but I believe that there is still a need for a slight additional revision of the grammatical and language used and the punctuation between the phrases and in the sentences used in the manuscript.

Also, you can use these references:

1. Bahadori, M., Hasanpoor, E., Yaghoubi, M. and HaghGoshyie, E., 2020. Determinants of a high-quality consultation in medical communications: a systematic review of qualitative and quantitative evidence. International Journal of Human Rights in Healthcare.

2. Hajebrahimi S, Janati A, Arab-Zozani M, Sokhanvar M, Haghgoshayie E, Siraneh Y, Bahadori M, Hasanpoor E. Medical visit time and predictors in health facilities: a mega systematic review and meta-analysis. International Journal of Human Rights in Healthcare. 2019 Nov 28.

3. Bahadori M, Yaghoubi M, Haghgoshyie E, Ghasemi M, Hasanpoor E. Patients’ and physicians’ perspectives and experiences on the quality of medical consultations: a qualitative study. JBI Evidence Implementation. 2020 Jun 1;18(2):247-55.

4. Hasanpoor E, Bahadori M, Yaghoubi M, Haghgoshayie E, Mahboub-Ahari A. Evidence-based management as a basis for evidence-based medical consultation. BMJ evidence-based medicine. 2020 Jun 1;25(3):83-4.

Response: 

Additional revision has been conducted to address the readability, grammar and language used. This also includes the punctuation between the phrases and in the sentences used in the manuscript.

This revision has been done so to achieve a closer alignment with the source references supplied by the reviewer.

---

## [Decision Letter · Decision Letter 2]

30 May 2022

Factors that influence older patients’ participation in clinical communication within developed country hospitals and GP Clinics: A systematic review of current literature.

PONE-D-21-29170R2

Dear Dr. Gaffney

We’re pleased to inform you that your manuscript has been judged scientifically suitable for publication and will be formally accepted for publication once it meets all outstanding technical requirements.

Kind regards,

Edris Hasanpoor

Academic Editor

PLOS ONE

---

## [Editor Report · Acceptance letter]

10 Jun 2022

PONE-D-21-29170R2 

Factors that influence older patients’ participation in clinical communication within developed country hospitals and GP Clinics: A systematic review of current literature. 

Dear Dr. Gaffney:

I'm pleased to inform you that your manuscript has been deemed suitable for publication in PLOS ONE. Congratulations! Your manuscript is now with our production department. 

Kind regards, 

on behalf of

Dr. Edris Hasanpoor 

Academic Editor

PLOS ONE